# Transporters, Ion Channels, and Junctional Proteins in Choroid Plexus Epithelial Cells

**DOI:** 10.3390/biomedicines12040708

**Published:** 2024-03-22

**Authors:** Masaki Ueno, Yoichi Chiba, Ryuta Murakami, Yumi Miyai, Koichi Matsumoto, Keiji Wakamatsu, Toshitaka Nakagawa, Genta Takebayashi, Naoya Uemura, Ken Yanase, Yuichi Ogino

**Affiliations:** 1Department of Pathology and Host Defense, Faculty of Medicine, Kagawa University, Kagawa 761-0793, Japan; chiba.yoichi@kagawa-u.ac.jp (Y.C.); murakami.ryuta@kagawa-u.ac.jp (R.M.); miyai.yumi@kagawa-u.ac.jp (Y.M.); matsumoto.koichi@kagawa-u.ac.jp (K.M.); s20d727@kagawa-u.ac.jp (K.W.); 2Division of Research Instrument and Equipment, Research Facility Center, Kagawa University, Kagawa 761-0793, Japan; nakagawa.toshitaka@kagawa-u.ac.jp; 3Department of Anesthesiology, Faculty of Medicine, Kagawa University, Kagawa 761-0793, Japan; takebayashi.genta@kagawa-u.ac.jp (G.T.); uemura.naoya@kagawa-u.ac.jp (N.U.); yanase.ken@kagawa-u.ac.jp (K.Y.); ogino.yuichi@kagawa-u.ac.jp (Y.O.)

**Keywords:** adherens junction, cerebrospinal fluid, choroid plexus, epithelial cell, tight junction, transporter

## Abstract

The choroid plexus (CP) plays significant roles in secreting cerebrospinal fluid (CSF) and forming circadian rhythms. A monolayer of epithelial cells with tight and adherens junctions of CP forms the blood–CSF barrier to control the movement of substances between the blood and ventricles, as microvessels in the stroma of CP have fenestrations in endothelial cells. CP epithelial cells are equipped with several kinds of transporters and ion channels to transport nutrient substances and secrete CSF. In addition, junctional components also contribute to CSF production as well as blood–CSF barrier formation. However, it remains unclear how junctional components as well as transporters and ion channels contribute to the pathogenesis of neurodegenerative disorders. In this manuscript, recent findings regarding the distribution and significance of transporters, ion channels, and junctional proteins in CP epithelial cells are introduced, and how changes in expression of their epithelial proteins contribute to the pathophysiology of brain disorders are reviewed.

## 1. Introduction

The brain restricts the entrance of solutes and toxic substances circulating in the blood by two barriers: The blood–brain barrier (BBB) and blood–cerebrospinal fluid (CSF) barrier (BCSFB) [1,2]. BBB consists of components, arranged in order from the inside, such as endothelial cells interconnected by tight junctions (TJs) with few vesicles and no fenestrations, two basement membranes, pericytes, and end-feet of astrocytes, providing a strong barrier function [1,3,4]. BBB endothelial cells are also equipped with various transporters, such as glucose transporter 1 (GLUT1/*SLC2A1*), to supply nutrient substances from the blood to the brain [2,3,5,6]. In this way, BBB endothelial cells have a significant carrier function as well as strong barrier function.

On the other hand, BCSFB is known to be present in a monolayer of epithelial cells of the choroid plexus (CP). CP protrudes into ventricles, is covered with epithelial cells with microvilli on the ventricle-facing (apical) side, and has an underlying basal lamina [7,8]. The stroma of CP is highly vascularized, and the endothelial cells of capillaries in the stroma are fenestrated, allowing the movement of intravascular substances into the CP stroma [7,9,10]. A monolayer of epithelial cells with TJs and adherens junctions (AJs) has a significant barrier function as BCSFB. Figure 1 shows the ultrastructure of CP epithelial cells (CPEs), with TJ/AJ on the lateral side and microvilli on the ventricle-facing side of the cytoplasm. CPEs also have a carrier function and are equipped with several kinds of transporters, including GLUT1 and ion channels, to transport nutrient substances, including glucose, and secrete CSF [7,10,11,12]. Standard concentrations of ionic compounds in CSF are reported as: Na^+^, 149 mM; K^+^, 2.9 mM; Cl^−^, 130 mM; HCO_3_^−^, 22 mM; showing 305 mOsm/L and pH 7.27, whereas the concentrations in interstitium of CP are Na^+^, 148 mM; K^+^, 4.3 mM; Cl^−^, 106 mM; HCO_3_^−^, 25 mM; showing 299 mOsm/L and pH 7.46 [7]. Many transporters and receptors are considered to contribute to CSF secretion in CPEs. Abnormalities in the transport of nutrient substances and CSF secretion due to CPE injuries may lead to brain dysfunction. In this manuscript, the relationship between abnormalities in transporters and several kinds of brain disorders is focused on. On the contrary, some papers indicate that CP may secrete only partial CSF [13] or may clear CSF [14,15]. In addition, Yamada and Mase reported that CSF is primarily produced as interstitial fluid (ISF) and drains into the subarachnoid space and ventricles as sinks, and that CSF produced by CP may have a special role in releasing hormones, cytokines, and other proteins and play a role in maintaining circadian rhythms and stress response homeostasis [16]. Therefore, the precise role of CP in CSF production remains controversial [17].

Some CSF flows in the brain, is mixed with ISF, and is considered to be excreted into the venous system. Much of CSF was previously thought to be reabsorbed directly through arachnoid granulations into the venous sinuses [18]. However, alternative pathways for CSF to return to the systemic circulation have been advocated to drain into cervical lymph nodes along peri- and paravascular spaces surrounding cerebral arterial vessels [19]. The two pathways are referred to as the intravascular periarterial drainage (IPAD) pathway [20,21] and glymphatic system [21,22]. As the two routes are necessary for the discharge of waste products produced in the brain, the disturbance of fluid flow through the two routes likely causes impaired brain function with increases or decreases in the amount of normal CSF compounds or metabolites and the accumulation of abnormal proteins in CSF. Glymphatic influx and clearance are more effective during sleep and can contribute to circadian rhythm formation in mice [23]. Accordingly, several kinds of abnormalities in CP, including the CP volume and pathological findings related to CSF secretion, are also reviewed.

## 2. Expressed Proteins in CPEs and Stroma

### 2.1. Expressed Proteins in CPEs

CPEs are characterized by the presence of some epithelial cytokeratins, vimentin, catenins, S-100 protein, podoplanin, transthyretin/prealbumin, and α1-antichymotrypsin [7,10,24]. Cytokeratin 8 (CK8), CK18, and possibly CK19 were reported to be intermediate filaments in CPEs [7,25]. However, CK19 has not been fully confirmed to be expressed in CPEs. The coexistence of CK8, CK18, vimentin, and S-100 protein in CPEs is unique and may contribute to the special functioning of CPEs.

CPEs are equipped with several transporters for CSF secretion in the apical and/or basolateral cytoplasmic membrane [7,26]. AQP1 (water channel), Na^+^-K^+^-ATPase, NKCC1 (Na^+^, K^+^, 2Cl^−^ cotransporter), NHE1 and NBCe2 (acid/base transporters), Clir and VRAC (Cl^−^ channels), and Kir7.1 and Kv (K^+^ channels) are expressed in the luminal membrane. Some acid/base transporters such as NBCn1 (Na^+^-dependent HCO_3_^−^ transporter), Ncbe (Na^+^-dependent Cl^−^/HCO_3_^−^ exchanger), and AE2 (anion exchange protein) are expressed in the basolateral membrane of CPEs. A large amount of AQP1 is expressed in the apical membrane of CPEs, whereas a small amount is expressed in the basolateral membrane [7].

### 2.2. Junctional Proteins Expressed between CPEs

The lateral intercellular space (LIS) between the lateral membranes of neighboring CPEs is generally narrow, and these cells were combined with TJ/AJ [7] (Figure 1). The TJs contain occludins, claudins, and the associated cytosolic ZO-1 [7]. Claudin-1, -2, -3, and -11 have all been demonstrated in TJs of CPEs [7,27]. On the other hand, zonula adherens, including cadherins and catenins, are expressed beneath TJs. Catenins are distributed along the lateral surface of CPEs [7]. Some cadherins have been reported to be expressed in AJs of CPEs [7]. P- and N-cadherins are known to be expressed in the lateral membrane and basal labyrinth of CPEs [7,28]. However, E-cadherin expression in CP was not clear until recently [7,28,29]. Takebayashi et al. [30] reported that mRNA and protein expression of E-cadherin was present in CP samples of 10-week-old mice, and immunoreactivity for E-cadherin was present in the lateral membrane of CPEs in the mice and aged human brains [30].

### 2.3. Proteins Expressed in CP Stroma

CP consists of CPEs with an underlying basal lamina and a stroma with connective tissues, including fenestrated capillaries [31,32] (Figure 1). Interestingly, the basal lamina of CPEs was reported to be continuous with the pial basal lamina [31,32]. The CP stroma in aged human brains was filled with fine fibrous tissues immunoreactive for collagen type 3, whereas calcified materials immunoreactive for collagen type 1 were occasionally present in the stroma [33].

Endothelial cells in CP capillaries have a special morphological feature, called fenestration, which enables intravascular molecules to invade the CP stroma. CD34, a transmembranous glycoprotein in endothelial cells, is well known as a representative endothelial cell marker. Not only CD34 but also several kinds of transporters, such as breast cancer resistance protein (BCRP/ABCG2), a urate transporter, were reported to be expressed in endothelial cells in the CP stroma as well as in endothelial cells in the BBB area [26,33,34] (Figure 2). Interestingly, immunoreactivity for GLUT1 is present in endothelial cells in the BBB area, whereas it is absent in endothelial cells in the CP stroma (Figure 2).

## 3. Localization of Several Kinds of Transporters in CPEs

### 3.1. Glucose Transporters

It is well known that GLUT1/*SLC2A1*, a representative glucose transporter in the brain, is predominantly expressed in the basolateral membrane of CPEs [1,35]. In addition, sodium/glucose cotransporter 2 (SGLT2/*SLC5A2*) was also reported in CPEs [26,36]. It was reported that the basal glucose level in brain interstitial fluid (ISF) of normoglycemic people showing a plasma glucose concentration of 6 mM is estimated to be 1.4 mM [37]. Accordingly, it is likely that these glucose transporters contribute to the transport of glucose in the stroma of CP into CSF via CPEs based on the concentration gradient. However, transporters on the apical membrane of CPEs remain unclear. The levels of pCO_2_ and HCO_3_^−^ in CSF were lower in the diabetic mellitus group than in the control group, whereas the concentrations of Na^+^ and Mg^2+^ in the blood were lower in the diabetic group than the control group [38]. Accordingly, some links between glucose transporters and ion channels in CPEs are suggested in patients with diabetes mellitus.

### 3.2. Fructose Transporters

Some experimental results using proteomic or transcriptome analysis, immunohistochemistry, Western blotting, and mRNA transcripts showed that GLUT5 (*SLC2A5*), a representative transporter for fructose, was expressed on the apical side of CPEs in mice, rats, and humans [39,40,41]. Expression of GLUT8 (*SLC2A8*), another fructose transporter, in CPEs was also confirmed by transcriptome and immunohistochemical analyses [39,42]. Interestingly, fructose transporters are expressed in CPEs but not in BBB endothelial cells, indicating the significance of the transport of fructose via CPEs between the blood and CSF. However, the physiological function of fructose in the brain as well as the direction of fructose transport between the blood and CSF remain to be clarified.

### 3.3. Urate Transporters

BCRP/*ABCG2*, GLUT9/*SLC2A9*, and urate transporter 1 (URAT1/*SLC22A12*) were reported to be urate transporters in BBB and/or BCSFB [26,34,43]. BCRP/ABCG2 is the main urate transporter on the luminal and apical membrane of the capillary endothelial cells of BBB and in CPEs, respectively [26,34]. The localization of ABCG2 in the luminal membrane of brain capillary endothelial cells suggests the transport of urate from the brain into blood. Immunohistochemical and in situ hybridization studies of GLUT9 and URAT1 in murine brains indicate the presence of GLUT9 in ependymal cells and brain capillaries, as well as the presence of URAT1 in ependymal cells [34]. In human brains, immunoreactivity for GLUT9 was present on the apical side of CPEs, whereas that for URAT1 was on the basal side of CPEs and also in ependymal cells at the third ventricular wall [43]. However, the mRNA and protein expression of URAT1 has not been confirmed in CPEs. In addition, mRNA of GLUT12, a transporter for urate and vitamin C [44,45], was expressed in CPEs of mice [45]. These findings suggest the possible movement of urate between CP and CSF.

### 3.4. Lactate Transporters

Monocarboxylate transporters (MCTs) catalyze the proton-linked transport of monocarboxylates, including L-lactate and pyruvate, in several kinds of cells [46,47]. Felmlee et al. [48] reviewed in detail the function, regulation, and role in health and disease of the MCT (*SLC16*) family. MCT1/*SLC16A1*, MCT2/*SLC16A7*, MCT3/*SLC16A8*, MCT4/*SLC16A3*, and MCT5/*SLC16A4* are known to be expressed as lactate transporters in CPEs of human and rat brains [25,41,46,49]. MCT1 and MCT2 were present on the apical side of CPEs in human brains, whereas MCT4 was present on the basal side [25,49]. The MCT3 gene was expressed in retinal pigments and CPEs of mice and rats [41,50]. Immunohistochemical localization of MCT3 was present on the basolateral membrane of CPEs in mice [50]. In addition, the mRNA of MCT3 was confirmed to be expressed in CPEs of rats [41,50].

### 3.5. Thyroid Hormone Transporters

MCT8 (*SLC16A2*) was established as an active transporter to transport thyroid hormones T3 and T4 [51]. It is now considered that MCT8, MCT10/*SLC16A10*, and organic anion transporting polypeptide 1C1 (OATP1C1) are specific thyroid hormone transporters, and that MCT8 and OATP1C1 are expressed in the brain [52]. Roberts et al. [53] reported that human MCT8 was immunohistochemically expressed in endothelial cells and also visible on apical and basal surfaces of CPEs, whereas immunoreactivity for OATP-14, known as OATP1C1, was present on both apical and basolateral surfaces of CPEs. On the other hand, mouse MCT8 was primarily expressed on the apical surface of CPEs, whereas OATP14 was present mainly on the basolateral surface of CPEs. We confirmed that immunoreactivity for MCT8 existed on the apical surface of human CPEs [25].

### 3.6. Iron-Regulatory Proteins and Iron Transporters

Iron is essential for the normal functioning of several kinds of cells. As excess iron can generate toxic reactive oxygen species, the metabolism of iron is tightly controlled. Astrocytes are largely responsible for regulating iron metabolism in the brain [54,55]. Concerning iron metabolism in the brain, iron in the blood crosses the BBB to enter the central nervous system mainly through transferrin and the transferrin receptor. Ionized iron in endothelial cells can be exported into the cerebral parenchyma by ferroportin, the only known iron export transporter. Extracellular iron is accumulated in astrocytes through the divalent metal transporter 1 (DMT1). Iron is partially stored in ferritin as ferric iron (Fe^3+^) or is exported from astrocytes through ion channels and ferroportin–ceruloplasmin cascade [55]. Under conditions of iron deprivation, astrocytes support neuronal iron uptake by releasing Fe^2+^ through ferroportin and secreting ceruloplasmin. Ceruloplasmin oxidizes Fe^2+^ to Fe^3+^, which binds to the transferrin receptor [55]. Hepcidin, which is synthesized by astrocytes and microglia, binds to ferroportin and induces its internalization and degradation [56,57]. In this way, the ferroportin–hepcidin system functions as the main pathway for cellular iron export and regulates cellular iron levels in astrocytes. Excess iron storage in astrocytes possibly results in oxidative damage to astrocytes, followed by neuronal cell injury. Hephaestin, a large membrane-anchored multicopper ferroxidase, is known to be involved in iron metabolism [58]. Dietary iron is exported across enterocytes by ferroportin, and hephaestin increases the efficiency of this process by oxidizing the transported iron to Fe^3+^ and promoting its release from ferroportin. Ferroportin, hepcidin, and hephaestin were reported to be present in human CPEs as well as astrocytes [59,60,61]. Interestingly, Ca^2+^ is required for the iron transport activity of human ferroportin, and the activity of ferroportin could be limited under conditions of hypocalcemia [58]. Accordingly, decreased concentrations of Ca^2+^ in CP may induce excess accumulation of Fe^2+^ in the cytoplasm of CPEs, possibly followed by the induction of oxidative cellular damage.

### 3.7. Ions and Water Transporters

Several kinds of transporters for Na^+^, K^+^, Cl^−^, HCO_3_^−^, and water are known to be expressed in the apical and/or basolateral membrane of CPEs [7,17,24,26,62,63]. As transporters expressed on the apical membrane of CPEs, Na^+^-K^+^-ATPase, NKCC1 (Na^+^, K^+^, 2Cl^−^ cotransporter), NHE1 and NBCe2 (acid/base transporters), Clir and VRAC (Cl^−^ channels), Kir7.1 and Kv (K^+^ channels), and AQP1 (water channel) are known. On the other hand, other acid/base transporters such as NBCn1 (Na^+^-dependent HCO_3_^−^ transporter), Ncbe (Na^+^-dependent Cl^−^/HCO_3_^−^ exchanger), and AE2 (anion exchange protein) are expressed in the basolateral membrane of CPEs. NKCC1 was reported by some studies to transport Na^+^, K^+^, and Cl^−^ from CPEs into the ventricle unidirectionally [7,62], whereas it was reported by others to transport these ions bidirectionally between CPEs and the ventricle [17,63]. In addition, NKCC1 is considered to contribute to the transport of water under some conditions [62]. A large amount of AQP1 necessary for water transport is expressed in the luminal membrane of CPEs, whereas a small amount is also known to be present in the basolateral membrane [7]. Transient receptor potential vanilloid type 4 (TRPV4), which was originally described as a calcium-permeable non-selective cation channel, is now recognized as a polymodal ionotropic receptor [64,65]. TRPV4, which is permeable to calcium, potassium, magnesium, and sodium, was reported to be present in the apical membrane of CPEs [17,66]. Accordingly, when TRPV4 is activated, Ca^2+^ can flow into CPEs [66]. These results indicate the contribution of TRPV4-mediated activities to transepithelial ion and water movement. Figure 3 shows the polarized distribution of representative transporters and ion channels in CPEs and the hypothesized directions of ions.

## 4. Localization of Transporters and Proteins in Junctions between Neighboring CPEs

There are the basal labyrinth (BL), lateral intercellular space (LIS), and a zone with TJ/AJ, which are aligned from the basolateral to apical sides, between the lateral membranes of neighboring CPEs [7]. Some junctional proteins function as adhesive molecules and contribute to CSF production.

### 4.1. Tight Junction

TJs are formed by occludin, claudins, and the associated cytosolic zonula occludens-1 situated along the entire lateral surface [7]. These are considered to regulate the function of TJs together. Occludin regulates the size-selective paracellular diffusion of hydrophilic molecules [67]. However, Saitou et al. [68] reported experimental findings suggesting a non-essential role for occludin in TJ formation. Claudin−1, −2, −3, and −11 were reported to be situated in TJs of CPE [27,69], whereas claudin-5 was identified as a critical regulator of BBB permeability in cerebral capillaries [70]. Interestingly, claudin-2 contributes to the transport of water as well as monovalent cations in the TJs of CPEs [71].

### 4.2. Adherens Junction

AJs are complexes that occur at cell-cell junctions and cell-matrix junctions in CPEs and are mainly composed of cadherins and catenins [7]. Cadherins are transmembranous proteins that have extracellular calcium ion-binding domains and depend on calcium ions to function. Adhesion between cells is mediated by extracellular cadherin domains. Cadherins are protected against degradation by proteases in the presence of calcium ions. Although the expression of cadherins in CP has been demonstrated, the specific forms of cadherins expressed remained unclear until recently [7,28,29]. Christensen et al. [28] reported the expression of P- and N-cadherins in the lateral membrane and basal labyrinth of CPEs. Recently, the mRNA and protein of E-cadherin were confirmed to be expressed in CP samples of 10-week-old mice using RT-PCR, Western blotting, and immunohistochemical analyses [30]. In addition, the presence of even or uneven expression of E-cadherin with expression of P- and N-cadherins was reported in the lateral membrane of CPEs in human brains [30]. Uneven expression of cadherins may be related to the epithelial-to-mesenchymal transition-like phenomenon and E-cadherin-to-P-cadherin switch, occurring in inflamed or injured tissues [72,73]. Calcium signaling is known to be pivotal to the circadian clock in the suprachiasmatic nucleus [74]. Accordingly, it is worth elucidating whether the expression of cadherins, which are calcium-dependent junctional proteins in CPEs, is related to circadian rhythm formation.

## 5. Alterations in CP Proteins with Aging and Brain Disorders

### 5.1. Age-Related Morphological Changes in CPEs

Some researchers have reported that CPEs show age-related morphological and functional changes. CPEs of elderly humans are known to exhibit decreases in the total volume, height, and length of apical villi, compared with cells of younger people [9,12]. Scarpetta et al. [75] reported age-related changes in murine CPEs, such as flattening of epithelial cells, reduction in microvilli length, an increase in interrupted TJs, and a decrease in mitochondrial density with elongation of mitochondria [75]. These morphological mitochondrial alterations were accompanied by increased superoxide levels and a decreased membrane potential [75]. Wakamatsu et al. [33] reported that immunoreactivities for osteopontin and collagen were present in the densely fibrous or calcified CP stroma of aged human brains. In the calcified stroma with psammoma bodies, the basal lamina immunopositive for type IV collagen was destroyed, and the covering CPEs were thin or disappeared [33]. In addition, Biondi ring tangles were present in the cytoplasm of CPEs in aging brains [76]. These tangles were specific intracellular inclusions in CPEs and were made of tightly packed bundles of long filaments with a diameter of around 10 nm, which were morphologically distinct from amyloid fibrils and paired filaments of neurofibrillary tangles in the case of Alzheimer’s disease (AD) [76]. It is not surprising that these morphological changes reported in aged brains cause age-related functional impairments of CP. Figure 4A shows a schematic illustration of the choroid plexus and ependymal cells. An illustration of normal-looking CPEs is shown on the left side of the figure, whereas CPEs in aged people are on the right side. On the left side, tight and adherens junctions between CPEs, the basement membrane, and fenestrated capillaries characterized by immunoreactivity for CD34 and ABCG2 but not GLUT1 are shown. On the other hand, thin epithelial cells, psammoma bodies, an injured basement membrane, and Biondi ring tangles seen in aged people are presented on the right side [33,76] (Figure 4A). Non-fenestrated capillaries in the BBB area show immunoreactivity for GLUT1 as well as CD34 and ABCG2 (Figure 2). Figure 4B–E presents hematoxylin and eosin stained images showing CPEs with a normal-looking appearance (B), thin or disappeared CPEs with dense fibrous materials or psammoma bodies in the stroma (C,D), and Biondi ring tangles in the cytoplasm of CPEs in human brains (E). Various morphological abnormalities in CP were reported in autopsied-aged brains [9,12,33,75,76].

### 5.2. CP Changes in Brain Disorders

A combination of increased CSF secretion caused by CP abnormalities and impaired CSF absorption likely induces posthemorrhagic hydrocephalus [77]. Liu et al. [78] reported on the relationship between abnormal CPEs and hydrocephalus or stroke. Periventricular white matter was injured with neutrophil infiltration into CP and white matter in a thrombin-induced hydrocephalus model [79]. Sadegh et al. [15] reported that overexpression of NKCC1, a bidirectional Na-K-Cl cotransporter, in CP accelerated CSF clearance and mitigated posthemorrhagic hydrocephalus, indicating a role of NKCC1 in CSF secretion.

Many studies have reported the contribution of CP abnormalities to the pathogenesis of neurodegenerative diseases. Senay et al. [80], using a novel magnetic resonance imaging-based segmentation method, reported a significant CP volume increase in early psychosis and a significant positive correlation between higher CP and higher lateral ventricle volumes in chronic psychosis, suggesting that CP enlargement may be a marker of an acute response around disease onset [80]. CP enlargement was significantly correlated with a higher allostatic load in patients with schizophrenia [81]. Histopathologically, an increased somal width of CPEs was present in the case of schizophrenia [82]. In AD patients, the CP volume was a good marker for the evaluation of tau deposition and neuroinflammation [83]. A larger CP volume was associated with the severity of cognitive impairment in the AD spectrum [84]. Increased CP volumes in AD also correlate with age and cognitive performance [85]. Using laser capture microdissection followed by label-free quantitative mass spectrometry of CP, signaling pathways in activated fatty acid beta-oxidation and inhibited glycolysis were changed in patients with AD compared with controls [86]. Quintela et al. [87] pointed out the roles of ions, glucose transporters, and transporters related to Aβ clearance in circadian regulation formation. Histopathologically, increased amyloid-β (Aβ) deposition, reduced TJ formation, and decreased expression of lipoprotein receptor-related protein 1 (LRP-1), a transporter for Aβ clearance, were reported in CPEs of AD patients [78]. These findings are considered to result in reduced CSF secretion, decreased Aβ clearance, and increased inflammatory cell invasion. The rhythmicity of clock genes was disrupted in CP of the APP/PS1 mouse model for AD [88]. Biondi ring tangles were present in the cytoplasm of CPEs in aged brains, especially AD brains [76]. The CP volume was associated with frontal or executive function, followed by the dementia conversion risk in patients with Parkinson disease (PD) [89]. The CP volume had the potential to serve as a biomarker for motor disabilities in PD patients [90]. Examination of the CP volume could assist in differentiating patients with frontotemporal lobar degeneration from healthy controls and in characterizing disease severity [91]. Butler et al. [92] reported that CP calcification may be a specific and relatively easily acquired biomarker of neuroinflammation and CP pathology in humans. Ricigliano et al. [93] reported using 3.0-T brain MRI that CPs are enlarged and inflamed in patients with multiple sclerosis (MS), particularly in those with relapsing-remitting MS with inflammatory profiles. CP enlargement was closely linked to emerging functional impairment, as depicted in mouse models and patients with MS [94]. In experimental autoimmune encephalomyelitis (EAE), the mouse model for MS, significant findings suggesting a relationship between the pathogenesis of MS and CP were found. It was previously reported that mucosal vascular addressin cell adhesion molecule 1 (MAdCAM-1) was upregulated in CPEs during EAE and likely facilitated the entry of leukocyte subsets into CP [95]. Recently, the deletion of MAdCAM-1 was confirmed to be relatively resistant to actively induced EAE [96]. These findings suggest a significant contribution of MAdCAM-1 expression in CPEs to the pathogenesis in mouse models and possibly also in patients with MS. Regarding the relationship between systemic infection and brain dysfunction, the CP-to-cortex network is considered to be involved in the spread of inflammation into the brain. Yang et al. [97] reported in human brains with COVID-19 that inflammation spread from CPEs to several kinds of brain cells with an increased number of CD68-positive macrophages, although SARS-CoV-2 could not be detected in the brain. In addition, *Helicobacter suis* DNA was previously reported to be more frequently found in gastric biopsies from patients with PD compared with a control group. *Helicobacter suis*, administered as bait in mice, was demonstrated to induce inflammation in the brain, including CP and cognitive decline [98]. In the mice, CP inflammation and disruption of BCSFB were detected, whereas the BBB in cerebral capillaries remained functional. It may be said that CP is a novel player in the stomach–brain axis [98]. These findings in MS models and systemic infection suggest that CP receiving blood supply from fenestrated capillaries may be a gateway to the spread of inflammation from systemic circulation into the brain. Accordingly, epithelial injury in CP with BCSFB dysfunction may make it more susceptible to the effects of systemic inflammation and easier to cause brain damage. Abnormal findings in CP of brains with physiological and pathologic disorders are introduced in Table 1.

## 6. Summary and Future Directions

Covering epithelial cells in CP are equipped with several kinds of transporters for ions and nutrient substances, whereas junctions between them are equipped with specific junctional proteins with barrier and carrier functions. These features make it possible to secrete CSF and supply nutrient substances to ventricles. In the stroma of CP, capillaries with fenestrated endothelial cells supply blood to CP, making it possible to transport a variety of nutritional substances into the CP stroma. On the contrary, it allows blood cells, including monocytes, to penetrate the stroma in the presence of pathological conditions such as inflammation. Penetrating inflammatory cells may affect several kinds of neuroglial cells in the brain, followed by the appearance or exacerbation of neurological symptoms. In addition, abnormal Ca^2+^ concentrations in CPEs may induce BCSFB dysfunction with degradation or switching of cadherins, oxidative cell damage by excess accumulation of Fe^2+^. It is required to clarify how abnormalities of transporters, ion channels, and junctional proteins in CPE due to CP injuries are involved in the pathogenesis of neurodegenerative disorders by analyzing the latest biological images of patients with the disorders and their autopsied brains.

## Figures and Tables

**Figure 1 biomedicines-12-00708-f001:**
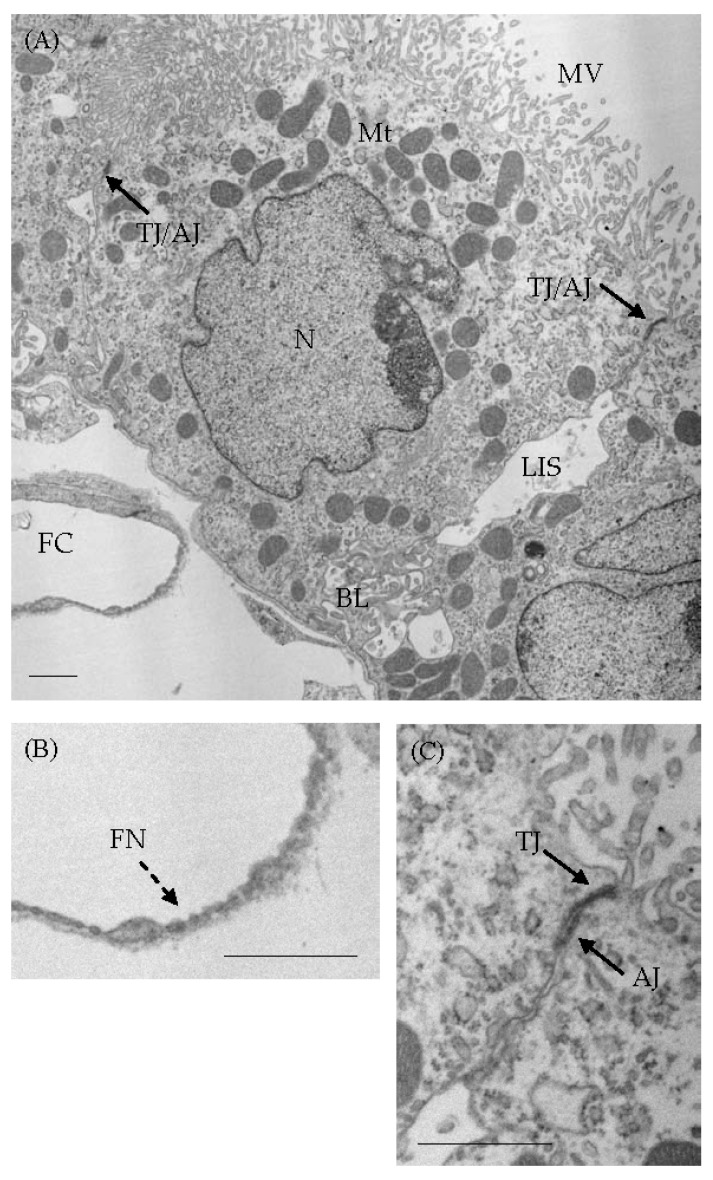
Representative electron microscopic images of epithelial cells from mouse CP tissues fixed with a mixed solution composed of paraformaldehyde and glutaraldehyde, followed by osmium tetroxide solution, and embedded in epoxy resin. (**A**) CPEs with junctions on the lateral side and microvilli on the apical side are seen. CPEs facing the ventricle are bound by tight and adherens junctions (indicated by arrows), whereas the lateral intercellular space (LIS) and basal labyrinth (BL) are present in junctional clefts. (**B**,**C**) Enlarged images show fenestrations (**B**: dotted arrow) in fenestrated endothelial cells and tight and adherens junctions (**C**: arrows). Scale bars indicate 1 μm. AJ: adherens junction; BL: basal labyrinth; FC: fenestrated capillar; FN: fenestration; LIS: lateral intercellular space; Mt: mitochondria; MV: microvilli; TJ: tight junction.

**Figure 2 biomedicines-12-00708-f002:**
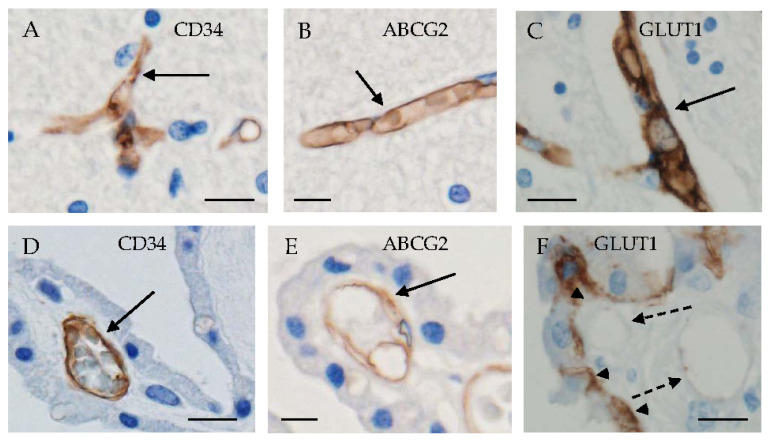
Representative immunoreactivity for CD34, ABCG2, and GLUT1 in microvessels of the brain parenchyma with BBB (**A**–**C**) and those in the CP stroma (**D**–**F**) of human brains. Arrows indicate immunoreactivity for CD34 (**A**,**D**), ABCG2 (**B**,**E**), and GLUT1 (**C**), whereas dotted arrows indicate no immunoreactivity for GLUT1 in the CP stroma (**F**). Arrowheads in (**F**) indicate positive immunostaining for GLUT1 on the basolateral surface of CPEs. These are compatible with findings reported in previous papers [24,26,33,34]. Scale bars indicate 10 μm.

**Figure 3 biomedicines-12-00708-f003:**
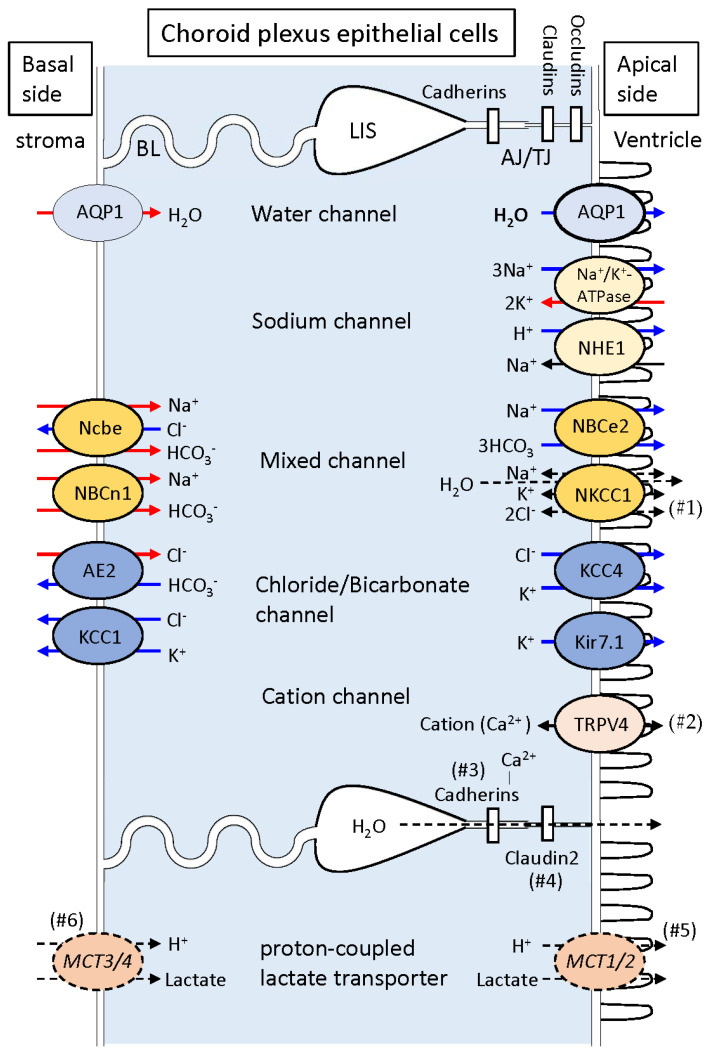
Polarized distribution of representative ion transporters/channels and proteins implicated in CSF secretion in the cytoplasmic membrane of CPEs. AQP1, a water channel, is expressed in large quantities in the luminal membrane, whereas it is also present less frequently in the basolateral membrane. On the apical side of the cytoplasmic membrane of CPEs, sodium potassium ATPase Na^+^,K^+^-ATPase, the sodium hydrogen exchanger 1 NHE1, sodium bicarbonate cotransporter e2 NBCe2, sodium potassium chloride cotransporter 1 NKCC1, potassium chloride cotransporter 4 KCC4, inward rectifying potassium channel Kir7.1, and transient receptor potential vanilloid TRPV4 (a non-selective cation channel) are expressed. On the basal side of the cytoplasmic membrane of CPEs, anion exchange protein AE2 (a chloride bicarbonate exchanger), potassium chloride cotransporter KCC1, Na^+^ dependent HCO_3_^−^ transporter NBCn1, and Na^+^-dependent Cl^−^/HCO_3_^−^ exchanger **Ncbe** are expressed. In the junctional space, claudin-2, a component of the tight junction, is involved in the permeation of water as well as monovalent cations, whereas cadherins, components of the adherens junction, have extracellular calcium ion-binding domains. Among MCTs, proton-coupled lactate transporters, MCT1 and MCT2 are immunohistochemically expressed on the apical side of CPEs, whereas MCT3 and MCT4 are considered to be on their basal side. (#1): Directions of Na^+^, K^+^, and Cl^−^ through NKCC1 were unidirectional (outward flow) according to some studies [7,62], whereas they were bidirectional according to others [17,63]. Steffensen et al. [62] reported that mouse CP has the ability to transport water against an osmotic gradient in a K^+^-induced, NKCC1-mediated manner [62]. (#2): TRPV4 may have a significant role in controlling ion and water flux. (#3): Cadherins have extracellular calcium ion-binding domains and depend on calcium ions to function. (#4): Claudin−2 likely contributes to the transport of water as well as monovalent cations in TJs. (#5, #6): Transmembranous directions of lactate and proton have not been determined at MCT1 and MCT2 on the apical side of CPEs or at MCT3 and MCT4 on the basal side of CPEs [25,41,49,50]. AJ/TJ: adherens and tight junctions; BL: basal labyrinth; LIS: lateral intercellular space.

**Figure 4 biomedicines-12-00708-f004:**
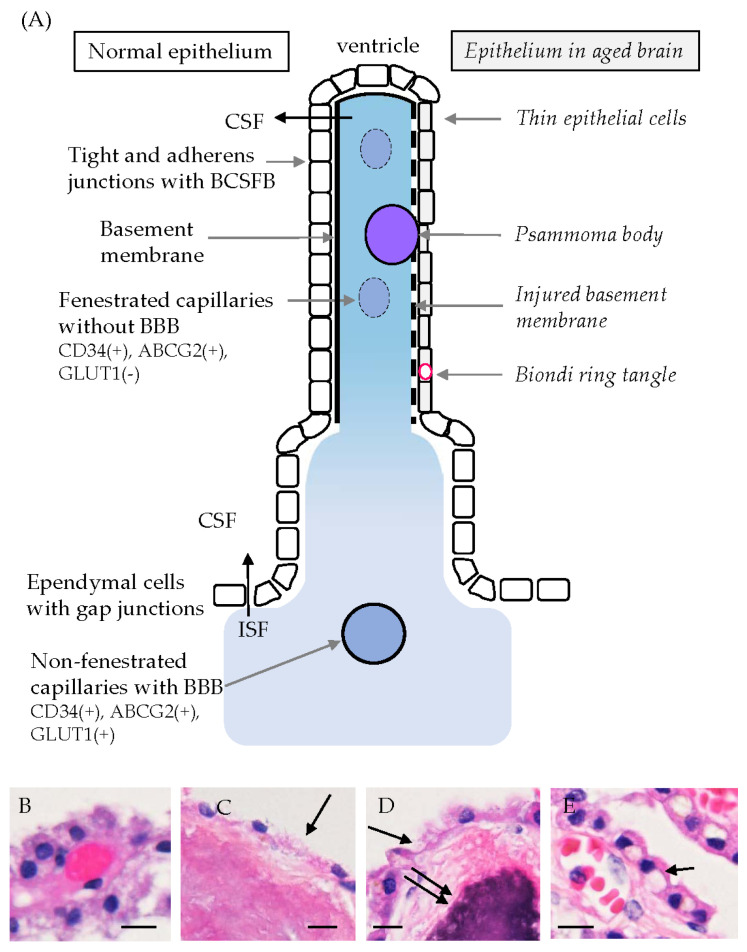
A schematic illustration of CP and ependymal cells in the normal (left side) and aged (right side) human brain (**A**), and images of CP stained with hematoxylin and eosin in human brains (**B**–**E**). (**A**) CPEs are bound by TJ/AJ, which are composed of BCSFB. Ependymal cells, mainly bound by gap junctions, are located between the ventricle and brain parenchyma. ISF can pass through the ependymal cell layer into the ventricle and is mixed with CSF. In aged human brains, psammoma bodies are frequently present in the stroma of CPEs, whereas Biondi ring tangles can be occasionally observed in the CP cytoplasm. Fenestrated capillaries without BBB in the CP stroma show CD34 (+), ABCG2 (+), and GLUT1 (−), whereas non-fenestrated capillaries with BBB in the BBB area show CD34 (+), ABCG2 (+), and GLUT1 (+). (**B**–**E**) Images stained with hematoxylin and eosin in CPEs of human brains of male patients in their 40 s (**B**), 80 s (**C**), and 70 s (**D**,**E**) are shown. Thick epithelial cells exhibit a normal-looking morphology, and the stroma is filled with capillaries (**B**). Epithelial cells (indicated by a long arrow) covering the fibrous stroma are very thin or have disappeared (**C**). Epithelial cells (indicated by a single arrow) covering the psammoma body (indicated by double arrows) are very thin or have disappeared (**D**). Biondi ring tangles (indicated by a short arrow) are seen in the cytoplasm of CPEs (**E**). Scale bars indicate 10 μm.

**Table 1 biomedicines-12-00708-t001:** Summary of abnormal findings reported in CP in presence of physiological and pathologic brain disorders.

[1] Morphometry of human brains by imaging techniques		
Psychosis	An increase in CP volume in early psychosis and a positive correlation between higher CP and higher lateral ventricle volumes in chronic psychosis.	[80]
Schizophrenia	CP enlargement and allostatic load.	[81]
Alzheimer’s disease	The CP volume is a good marker for the evaluation of tau deposition and neuroinflammation.	[83]
	A larger CP volume is associated with the severity of cognitive impairment in the AD spectrum.	[84]
	Increased CP volumes in AD correlate with age and cognitive performance.	[85]
Parkinson’s disease	The CP volume is associated with frontal or executive function, followed by the dementia conversion risk.	[89]
	The CP volume has the potential to serve as a biomarker of motor disabilities.	[90]
FTLD	The CP volume can assist in differentiating patients with FTLD from healthy controls and characterizing disease severity.	[91]
Multiple sclerosis	CP enlargement and inflammation	[93]
	CP enlargement is closely linked to emerging functional impairment.	[94]
[2] Pathological findings of CPEs and interstitium by histological or molecular biological techniques		
Hydrocephalus	Overexpression of NKCC1 mitigates posthemorrhagic hydrocephalus.	[15]
	Increased CSF secretion and impaired CSF absorption in the posthemorrhagic state.	[77]
	The relationship between abnormal CPEs and hydrocephalus or stroke.	[78]
	Periventricular white matter injury with neutrophil infiltration into CP and white matter in thrombin-induced hydrocephalus.	[79]
Schizophrenia	Increased somal width of CPEs.	[82]
Alzheimer’s disease	Biondi ring tangles.	[76]
	Increased amyloid-β deposition, reduced TJ formation, and decreased expression of LRP-1.	[78]
	Changes in signaling pathways associated with cell metabolism including activated fatty acid beta-oxidation and inhibited glycolysis.	[86]
Multiple sclerosis	A large number of HLA-DR immunostained T lymphocytes in CPEs.	[78]
Aging, inflammation, or others	Decreases in total volume, height, and length of microvilli of CPEs in the elderly.	[9,12]
	The basement membrane immunopositive for type IV collagen is destroyed and covering CPEs are thin or have disappeared.	[33]
	Age-related changes in flattening of CPEs, reduction in microvilli length, an increase in interrupted tight junctions, and a decrease in mitochondrial density with elongation of mitochondria of mice.	[75]
	Biondi ring tangles are present in aged brains.	[76]
	Roles of Na+/K+-ATPase, GLUT1, and transporters related to Aβ clearance in circadian regulation in CPEs.	[87]
	CP calcification may be a specific and relatively easily acquired biomarker of neuroinflammation and CP pathology in humans.	[92]
	MAdCAM-1 is upregulated in CPEs during experimental autoimmune encephalitis and may facilitate the entry of leukocyte subsets into CP.	[95]
	In human brains with COVID-19, barrier cells of the CP sense and relay inflammation into the brain with infiltration of increased number of CD68-positive macrophages into the stroma of CP.	[97]
	*Helicobacter suis* infection induces brain inflammation associated with cognitive decline, including CP inflammation, and the CP is a novel player in the stomach–brain axis.	[98]

AD: Alzheimer’s disease, FTLD: frontotemporal lobe degeneration.

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
