# Peer review of "Transporters, Ion Channels, and Junctional Proteins in Choroid Plexus Epithelial Cells"

_biomedicines, 2024, doi:10.3390/biomedicines12040708_

Round 1

Reviewer 1 Report

Comments and Suggestions for Authors

It is a very good review that summarizes the characteristics of the epithelial cells of the choroid plexus and the changes they undergo with age and their relationship with different degenerative diseases.

I make the following observations:

- Check Figure 2 to make sure the references match the information in the figure caption.

- Review the last 2 references in table 1.

- I suggest updating the references and information in point 3.6, on the participation of astrocytes in iron metabolism.

Author Response

To Reviewer 1:

It is a very good review that summarizes the characteristics of the epithelial cells of the choroid plexus and the changes they undergo with age and their relationship with different degenerative diseases.

I make the following observations:

- Check Figure 2 to make sure the references match the information in the figure caption.

To the comment: Thank you for your comment. I changed one sentence including reference numbers in the figure caption of Fg. 2.

- Review the last 2 references in table 1.

To the comment: I changed descriptions of the last 2 references [97, 98] in Table 1 and added descriptions in Discussion..

- I suggest updating the references and information in point 3.6, on the participation of astrocytes in iron metabolism.

To the comment: According to the comment, I deleted one reference reported by Dringen et al. (2007) and newly added one reference reported by Li et al. (2024) in the reference [55]. A paper [55] reported by Miyajima in the original manuscript was moved to [54] in the revised manuscript. In addition, to update information on iron metabolism in astrocytes, I deleted 3 sentences and added 6 sentences in point 3.6 of the revised manuscript.

Reviewer 2 Report

Comments and Suggestions for Authors

The review describes immunohistochemical reactions to reveal the presence and position of ion channela, transporters and junctional proteins in choroid plexus epithelial cells. The authors are experienced in the field and are combining the results of their own laburatory work, which they already exploited for many articles,  with the presentation and discussion of well chosen articles in the field.

The article is well written, the illustrations are excellent and the literature cited is - according to the experience of the group - comprehensive including recent papers. 

The relation of the results presented to the circadian rhythm and its disturbances is hard to follow. The last sentence of the summary of the desirable development of highly sensitive imagine techniques to examine CP abnormalities is not a good conclusion to summarize the more stochastic results of MRI imaging presented in the preceding pages. The strength of the paper is the sophisticated presentation and discussion of the immunohistochemistry. The MRI part does not reach the same quality and even can be deleted, if the authors want to focus on the topic of their headline: transporters, ion channels and junctional proteins. Neither of those can be shown by MRI.

Comments on the Quality of English Language

Well written.

Author Response

To Reviewer 2:

The review describes immunohistochemical reactions to reveal the presence and position of ion channela, transporters and junctional proteins in choroid plexus epithelial cells. The authors are experienced in the field and are combining the results of their own laburatory work, which they already exploited for many articles,  with the presentation and discussion of well chosen articles in the field.

The article is well written, the illustrations are excellent and the literature cited is - according to the experience of the group - comprehensive including recent papers. 

The relation of the results presented to the circadian rhythm and its disturbances is hard to follow.

To the comment: A sentence “ However, it remains unclear how junctional components contribute to circadian rhythm formation related to calcium signaling” was changed to “However, it remains unclear how junctional components as well as transporters and ion channels contribute to the pathogenesis of neurodegenerative disorders”. In addition, we deleted a phrase “circadian rhythm dysfunction” from the sentence before the last in point 6. Summary and future directions.

The last sentence of the summary of the desirable development of highly sensitive imagine techniques to examine CP abnormalities is not a good conclusion to summarize the more stochastic results of MRI imaging presented in the preceding pages.

To the comment: We deleted the last sentence of the summary. On behalf of the sentence, a new sentence was inserted in the last of the summary.

The strength of the paper is the sophisticated presentation and discussion of the immunohistochemistry. The MRI part does not reach the same quality and even can be deleted, if the authors want to focus on the topic of their headline: transporters, ion channels and junctional proteins. Neither of those can be shown by MRI.

To the comments: I can understand the reviewer’s reasonable comments. However, analysis of biological images before death as well as pathological findings of CP is required to clarify the involvement of CP to the pathogenesis of neurodegenerative disorders, although it is not running at this time. We think it is important to compare the two findings. Accordingly, we present the biological image findings as well as pathological findings of neurodegenerative disorders in the review paper.